

# Psychophysical measurements in children: challenges, pitfalls, and considerations

Caroline Witton, Joel B. Talcott and G. Bruce Henning

Aston Brain Centre, Aston University, Birmingham, United Kingdom

## ABSTRACT

Measuring sensory sensitivity is important in studying development and developmental disorders. However, with children, there is a need to balance reliable but lengthy sensory tasks with the child's ability to maintain motivation and vigilance. We used simulations to explore the problems associated with shortening adaptive psychophysical procedures, and suggest how these problems might be addressed. We quantify how adaptive procedures with too few reversals can over-estimate thresholds, introduce substantial measurement error, and make estimates of individual thresholds less reliable. The associated measurement error also obscures group differences. Adaptive procedures with children should therefore use as many reversals as possible, to reduce the effects of both Type 1 and Type 2 errors. Differences in response consistency, resulting from lapses in attention, further increase the over-estimation of threshold. Comparisons between data from individuals who may differ in lapse rate are therefore problematic, but measures to estimate and account for lapse rates in analyses may mitigate this problem.

# INTRODUCTION

Sensory processing in hearing and vision underlies the development of many social and cognitive functions. Consequently, accurate measurement of sensory function has become an important component of research into human development—particularly atypical development. Subtle differences in sensory sensitivity, especially for hearing, have been associated with a range of disorders diagnosed in childhood, e.g., dyslexia (*Benassi et al., 2010*; *Habib, 2000*; *Hämäläinen, Salminen & Leppänen, 2013*), specific language impairment (SLI) (*Webster & Shevell, 2004*), and autism (*O'Connor, 2012*; *Simmons et al., 2009*). The potential significance of sensory impairments, either as causal or as associated factors in such disorders, provides strong motivation for measuring sensory processing during development.

However, the psychophysical methods often used to estimate sensitivity present some key challenges in working with children (*Wightman & Allen, 1992*) because the most reliable methods require both good attention and short-term memory skills. Such cognitive demands are particularly acute where stimuli are presented sequentially, as in most auditory

Corresponding author
Caroline Witton,
c.witton@aston.ac.uk

experiments. Several authors have noted that children often respond erratically in these tasks. For example, 41% of children with dyslexia or SLI who completed up to 140 runs of an auditory frequency-discrimination task responded inconsistently with no improvement across runs (*McArthur & Hogben, 2012*). Other studies have confirmed this observation, i.e., that nearly 50% of children may be unable to produce response-patterns with adult-like consistency even after training (*Halliday et al., 2008*). Inconsistent responding produces widely varying scores on psychophysical tasks (*Roach, Edwards & Hogben, 2004*) and reported scores can be seriously misleading. Figure 2 of the paper by *McArthur & Hogben (2012)* clearly illustrates the widely differing kinds of performance observed in the staircase tasks used with children in their study. Extreme instability of response patterns was independent of the overall sensitivity level of individual children, and occurred both in children with apparently average sensitivity and those with lower sensitivity.

Moreover, even typically developing children often have difficulty concentrating on simple psychophysical tasks. In our study of frequency discrimination in 350 school children aged from 7–12 years, we found that children incorrectly identified 19% ($\pm 1$ SD of 15%) easy "catch trials" (*Talcott et al., 2002*). This percentage is much higher than the 5% "lapse rates" that are typically found in trained-adult psychophysical observers (*Wichmann & Hill, 2001a*; *Wichmann & Hill, 2001b*). Children's reduced performance compared to adults may stem from several factors, such as personal motivation, ability to consistently operationalise stimulus dimensions like pitch or duration, the ability to direct attention to a single stimulus dimension, and the ability to maintain vigilance. The integrity of these factors can be particularly impaired in children with developmental disorders (e.g., *Welsh, Pennington & Groisser, 1991*; *Karmiloff-Smith, 1998*; *Thomas et al., 2009*; *Cornish & Wilding, 2010*). Further, in studies where data are collected outside of the laboratory, the testing environment may be a classroom or even a corridor—so distractions are difficult to control and will interact differentially with these intrinsic developmental factors, possibly resulting in quite unstable performance.

The aim of this study was to use simulations to explore some of the factors that constrain the interpretation of sensory data acquired using adaptive psychophysical procedures in neurodevelopmental research. Some of the problems we discuss are relevant to many types of psychophysical experiment, but they are particularly acute in the interpretation of the results obtained from staircase procedures where a single measure, the "threshold", is used to represent an individual's sensitivity or performance. In work with (1) trained responders whose response patterns are highly stable, and (2) tasks with well understood underlying psychophysical properties, adaptive procedures can offer a quick and reliable estimate of threshold. However staircases depend on the assumption that the stability, shape, and slope of the underlying psychometric function are equivalent across participants, which can be problematic in the context of developmental research. For example, there is evidence that the slope of the psychometric function relating performance to the variable under study can change with developmental age (e.g., *Buss, Hall & Grose, 2009*). In many developmental studies, new tasks are used before the details of the underlying function have been determined in trained adults, and little is known about behaviour in an untrained or paediatric population.

| Table 1 | Key terminology. Definitions of key terms used in the text. |
|---|---|
| Psychometric function | The relation between stimulus level and the proportion of correct responses made by the participant. |
| Underlying psychometric function | The veridical relation between stimulus level and the probability of a correct response as used in a model for predicting a participant's psychometric function. In behavioural data, either assumed or inferred from a measured psychometric function. |
| Stimulus Level | A measure of the stimulus characteristic being manipulated by the experimenter. E.g., frequency difference, gap width. |
| 2-alternative forced-choice (2AFC) | A commonly-used psychophysical task design, in which two stimuli are presented on every trial and the participant judges which of the two is the 'target'. |
| Threshold | Often defined as the stimulus level at which the subject correctly identifies the target interval at some level of performance, usually 75% correct in a 2AFC procedure. |
| Adaptive procedure or 'staircase' | A method for estimating threshold by adjusting stimulus levels from trial-to-trial until a stopping-rule is reached. |
| Reversal | A reversal occurs when, in an adaptive procedure, a sequence of stimulus level adjustments that have been all in one direction (e.g., all to smaller levels) changes direction. |
| Stopping rule | The condition required to terminate an adaptive procedure; often a fixed even number of reversals but occasionally, where step sizes change, a given small step size. |
| Lapse rate | The proportion of trials upon which the participant fails to respond or responds randomly to the stimulus. Impossible to measure but can be estimated. It is often assumed that the lapse rate is independent of stimulus level. |

## Psychophysical methods

Sensory sensitivity is best measured using psychophysical tasks based on the principles of signal detection theory (*Green & Swets, 1966*). Table 1 provides definitions for some key terms used below. The most commonly used design is a 2-alternative, forced-choice (2 AFC) paradigm, where, in studies of hearing, participants are asked to listen to a number of trials in which pairs of stimuli are presented in two separate "observation intervals". Participants are required to report which interval contained one of the stimuli (the "target" stimulus). For example, the task might be auditory frequency discrimination in which case the intervals might contain tones that differ only in frequency. The participant would be required to select the interval with the higher-frequency tone—the target. The size of the frequency difference would be manipulated by the experimenter. Or, in a gap-detection experiment, each interval might contain a burst of noise, only one of which, the target, contains a short interval of silence. The participant would be required to pick the interval containing the silent gap, with the duration of the gap manipulated by the experimenter. In all cases, the target is as likely to be in the first as in the second interval.

By presenting an extensive series of trials, the experimenter seeks to determine the relation between the proportion of correct responses and values of the manipulated parameter; we will call those values the "stimulus level".
a.

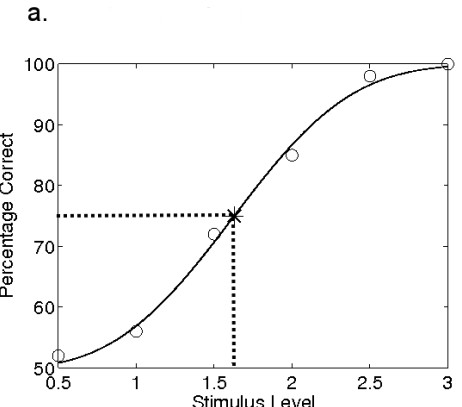

b.

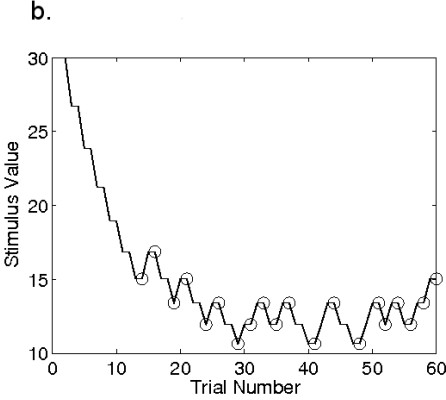

**Figure 1  Hypothetical psychophysical data.** Data from a hypothetical psychometric function (A) and an adaptive procedure-track (B). In (A), the data showing percentage of correct responses for six stimulus values have been fit with a Weibull function; dashed lines show the intersection of threshold and the 75%-correct point on this function. In (B), the procedure terminates after 20 reversals, indicated by circles.

Figure 1A shows hypothetical, idealised data illustrating a typical 2 AFC experiment. The relation between stimulus level and the probability of correctly identifying the target interval, known as the "psychometric function", is estimated by making a number of observations at different stimulus levels, marked by circles here. The data reflect an unknown "underlying psychometric function" that is the *true* relation between stimulus level and the probability of a correct response. Although other shapes including nonmonotonicity are possible, with trained observers the psychometric function typically follows a sigmoidal shape and is often fitted with a smooth curve, such as the Weibull function, as shown in Fig. 1A (*Macmillan & Creelman, 1991*). Fitting enables interpolation between the stimulus levels used, and hence determination of a "threshold" corresponding to some performance level. In Fig. 1A, for example, threshold is defined as a performance level of 75% correct, corresponding to a stimulus level of 1.6. Ideally, the only source of variability is the binomial variability in the number of correct responses at each stimulus level, though children and even trained adults rarely respond as consistently as this (*Talcott et al., 2002*).

In studies of sensory sensitivity, threshold is typically the key parameter of interest. However, it is desirable to measure the entire psychometric function for the additional information in the slope of the function. For example, several points on the function might well be needed in comparisons across conditions if the underlying functions are not parallel. An accurately-measured psychometric function is about the best that can be done in quantifying any sensory capability.

Unfortunately, a large number of trials are needed to estimate a full psychometric function accurately: perhaps more than 100 trials at each of at least five appropriately placed stimulus values (*Wichmann & Hill, 2001a*). When the listener is a young child, or an untrained or poorly motivated adult, it can be difficult to ensure their engagement with any task for so many trials. Researchers therefore frequently use an adaptive procedure (or "staircase") to reduce the number of trials. Adaptive methods typically attempt to estimate
just one point on the psychometric function—the threshold. Stimulus values are adjusted from trial-to-trial so that the stimulus level, it is hoped, eventually settles near the desired point on the underlying psychometric function. The change in stimulus level from trial to trial (the "step size", which can be fixed or variable) depends on the subject's responses in preceding trials via an "adjustment rule". For example, in a two-down, one-up staircase with fixed 1-dB steps (*Levitt, 1971*), the stimulus level (for e.g., gap duration) would be divided by a factor of 1.122 (a reduction of 0.05 log units or 1 dB) following two successive correct answers, and increased by the same factor after each incorrect response.

A change in the direction of the progression of the stimulus level is called a "reversal". To illustrate, Fig. 1B shows a simulated adaptive procedure with a 1dB step size and the same underlying psychometric function as in Fig. 1A. In Fig. 1B, the stimulus level is shown as a function of the trial number and the reversals are circled. The procedure ends when it fulfils the "stopping rule", which might be defined by a certain number of reversals. Alternatively, the stopping rule might be met when a criterion (small) step size has been reached. An individual's threshold is then calculated as the average stimulus level across an even number of reversal points at the end of the procedure.

In an adaptive procedure, the exact point on the psychometric function towards which a staircase asymptotically converges depends on the adjustment rule: in the 2-up 1-down staircase example in Fig. 1B, the asymptotic performance level is 70.7% correct (*Levitt, 1971*). The asymptotic performance level, the rate at which the staircases converges, and the accuracy of the estimated threshold all depend on the stopping rule, the step size and its adjustments, the response consistency of the subject, and the unknown slope of the psychometric function underlying the subject's performance.

## Limitations of adaptive procedures in paediatric and clinical settings

Adaptive procedures have the advantage of reducing the number of trials needed to estimate the threshold. However these procedures typically assume that the underlying psychometric function is stable both in slope and in threshold over successive trials (*Leek, Hanna & Marshall, 1991*; *Leek, Hanna & Marshall, 1992*; *Leek, 2001*). Indeed, the majority of adaptive procedures were developed for use in laboratory settings, where the psychophysical properties of the stimulus under investigation are well-understood, and the subjects are both highly trained and highly motivated. Under these conditions, the asymptotic behaviour of adaptive procedures with large numbers of reversals, and consistent response-patterns, are well-understood. Further, with a known underlying function, step sizes can be optimized to produce rapid convergence to the stimulus level that corresponds to the desired level of performance. However, these conditions are rarely met in studies with children or untrained subjects where, because of time-constraints, staircases are often stopped after a relatively small number of reversals and the underlying psychometric function is poorly described. Comparatively little is known about the performance of adaptive procedures under these conditions. Further to this, even motivated observers occasionally lose concentration, make impulsive motor responses, or fail to respond. Such "lapses" (other than failures to respond), have previously been modelled as trials where the response in the 2AFC paradigm is correct with probability of 0.5 irrespective of the
stimulus level. Lapse rates for trained adults rarely approach 5% (*Wichmann & Hill, 2001a*; *Wichmann & Hill, 2001b*), but much higher lapse rates are realistic for children (e.g., 19% ± 1 SD of 15% in *Talcott et al. (2002)*).

In this study, we aimed to characterise some of the properties of adaptive procedures that are particularly relevant for studies with young subjects. It was not our intention to explore the intricacies of the many adaptive procedures in use: their asymptotic behaviour has been carefully studied and the effects of different stopping rules and step sizes thoroughly explored (see *Leek, 2001*, for review). Instead, our aim was to use simulations to determine how a commonly adopted adaptive procedure performs under realistic paediatric experimental conditions; how efficiency changes when used with small numbers of reversals; and how the (usually unknown) underlying psychometric function affects threshold estimation and the use of thresholds in statistical analyses.

Specifically, we explored the effects of the varying adaptive procedure parameters, specifically the number of reversals and adjustment rule; and varying the participant characteristics of psychometric function slope, veridical threshold, and lapse rate. Our first objective was to determine how these factors affect the accuracy of threshold estimation in individual staircase measurements. Our second objective was to determine how these factors may affect the statistics of datasets containing thresholds for groups of participants, with particular reference to the kind of group comparisons that are common in studies of developmental disorders.

## METHODS

Adaptive procedures (staircases) were simulated using model participants with a known (veridical) underlying psychometric function described by a cumulative Weibull function (the smooth curve in Fig. 1A and Eq. (1)), using Matlab software (The Mathworks Inc., Natick, MA, USA). A formulation of the Weibull function giving the probability, $p(x)$, of correctly indicating the signal interval at any given stimulus level is:

[1] Theoretical convergence points determined by the adjustment rule of a staircase are based on the assumption of a cumulative normal psychometric function. When simulated using a Weibull function, the procedures converge at a very slightly lower value; the 2-down, 1-up staircase converges at 70.2% correct rather than 70.7% after 1,000 reversals. These small differences are negligible in the context of the effects described here.

$$p(x) = 1 - (1-g)\exp\left(-\left(k\frac{x}{t}\right)^{\beta}\right), \tag{1}$$

where $x$ is the stimulus level, $t$ is the threshold (i.e., the stimulus level at the theoretical convergence point of the adaptive procedure; e.g., $t = 10$ for performance converging asymptotically at 70.7% correct in Fig. 2A),[1] $\beta$ determines the slope of the psychometric function, $g$ is the probability of being correct at chance performance (0.5 for a 2 AFC task), and $k$ is given by:

$$k = \left(-\log\left(\frac{1-c}{1-g}\right)\right)^{\frac{1}{\beta}}. \tag{2}$$

The parameter $c$ is determined by the tracking rule of the staircase—it corresponds with the point at which the procedure will theoretically converge, for example on 70.7% for our 2-down, 1-up staircase. The slope parameter, $\beta$, is usually unknown but it is fixed in each of our simulations.

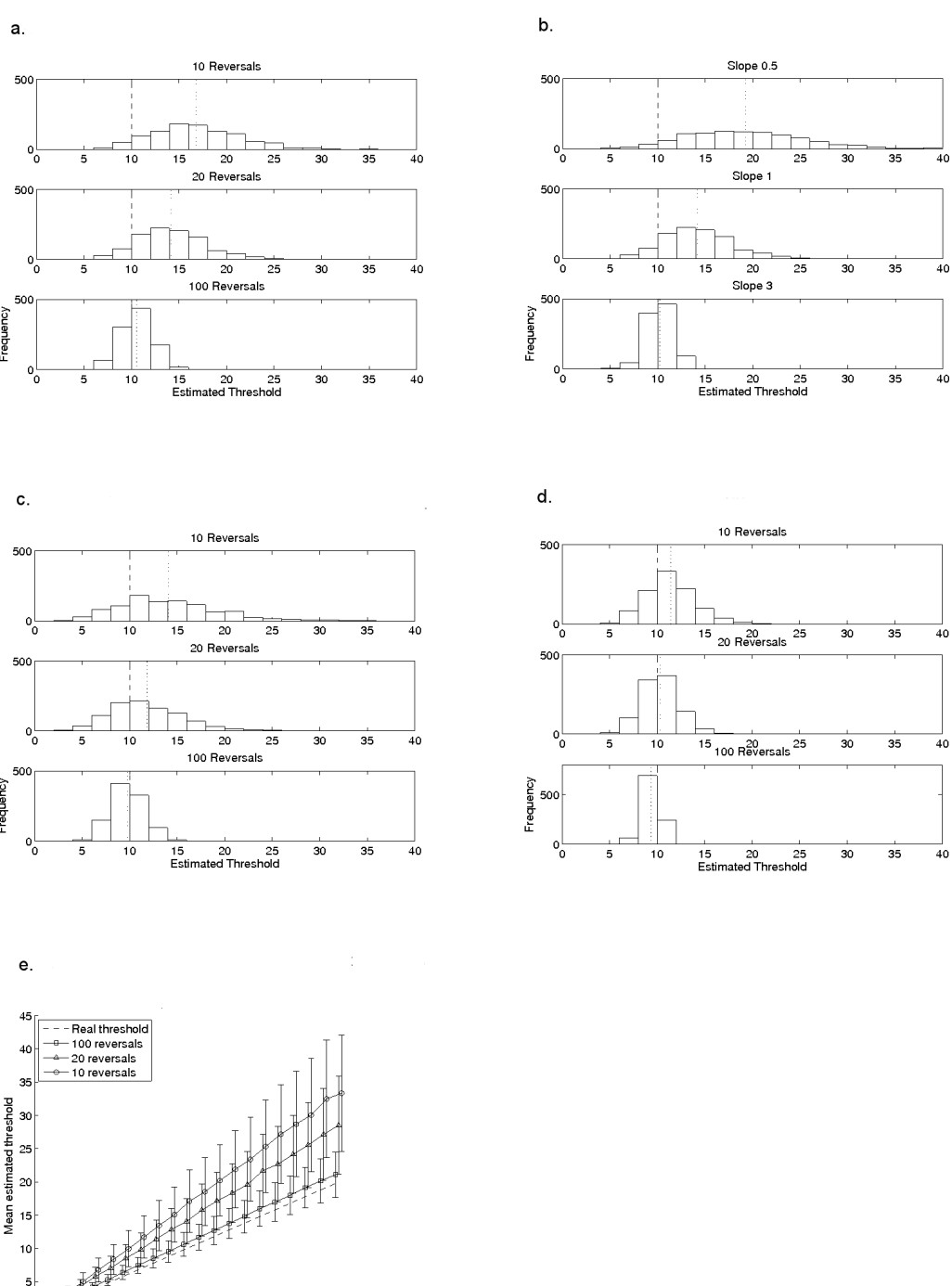

**Figure 2 Effects of reversal-count, slope, step-size, and adjustment rule on a typical staircase procedure.** The effects of reversal count (2A), slope (2B) and step-size (2C) on the mean and variability of thresholds measured with a 2-down, 1-up procedure. In all plots, the model subject had a known and fixed threshold of 10, indicated by the dashed line; the dotted line indicates the mean of the estimated thresholds. In (A), data are shown for 10, 20 and 30 reversals when the 
**Figure 2 (…continued)**
model subject had a fixed slope ($\beta$) of 1, for a 2-down, 1-up (1 dB) adaptive procedure. In (B), data are for 20 reversals with the same 2-down, 1-up procedure but the value of $\beta$ is either 0.5, 1, or 3. In (C), all parameters are the same as in (A) but the step-size of the adaptive procedure is 2 dB instead of 1 dB. (D) illustrates the different relationship with reversal-count when the adjustment rule is changed, in this case to a 3-down, 1-up (1 dB) procedure. (E) shows mean thresholds, estimated by the 2-down, 1-up (1 dB) adaptive procedure, for a set of model subjects with a range of thresholds between 1 and 20 ($\beta = 1$). Their real thresholds are plotted against mean estimated thresholds based on 10, 20 and 100 reversals. The error bars indicate $\pm 1$ standard deviation in the estimated threshold. Points are artificially offset from each other to facilitate interpretation of the error bars.

Each simulation commenced with a stimulus level set to 3 times the model subject's (known) threshold. The stimulus value was adjusted trial-by-trial according to the model's responses and the adjustment rule and step size of the staircase. For example, for a stimulus level corresponding to 80% correct on the underlying veridical psychometric function, the model subject would have an 80% probability of responding correctly on every trial in which that stimulus level was used in the simulation. Distributions of threshold estimates were produced using 1,000 simulations of the given adaptive procedure with the same step size, stopping rule, and mode of estimating the threshold. We used 1,000 simulations because pilot testing showed that this number produced stable results. Analyses of the effects of the number of reversals used to estimate the threshold, the adjustment rule, and response consistency were then undertaken.

For the simulations of threshold estimation under normal conditions (i.e., stable responding), the model participant always had a threshold $t = 10$, and unless otherwise specified, slope $\beta = 1$. For the majority of simulations, a 2-down, 1-up staircase with 1-dB steps was used (*Levitt, 1971*). The effects of the stopping rule (i.e., the number of reversals to finish) were explored, for 10, 20 and 100 reversals—chosen because 10 or 20 reversals are commonly used in the literature on sensory processing in developmental disorders such as dyslexia, for example. One hundred reversals exceeds the number typically used even in detailed psychophysical studies of trained adults. Also explored were the effects of the procedure's step-size (2 dB or 1 dB), and its adjustment rule (2-down, 1-up or. 3-down, 1-up); and the slope of the model observer's veridical psychometric function ($\beta = 0.5$, 1, or 3). For comparison, in trained adult subjects, 2 AFC psychometric functions for frequency discrimination have a slope of approximately 1 (*Dai & Micheyl, 2011*) whereas gap detection has a steeper slope (*Green & Forrest, 1989*). See *Strasburger (2001)* for conversions between measures of slope.

To examine the effects of the number of reversals with varying thresholds in individual participants, the model participant had a slope $\beta = 1$, but the threshold for different participants varied between 1 and 20. Thresholds were then estimated using the 2-down, 1-up staircase with 1 dB steps. Mean estimated thresholds produced by the staircase were compared with the veridical thresholds of the model participants.

The effects of number of reversals on group comparisons were explored using groups of 1,000 model participants (all slope $\beta = 1$) with thresholds drawn from a known Gaussian distribution, centred on an integer value between 5 and 12. We chose a standard deviation that was 20% of the mean, since Weber's Law stipulates a standard deviation that is a

constant fraction of the mean. Thresholds were estimated using both the 2-down, 1-up staircase procedure with 1 dB steps, and a 3-down, 1-up procedure with 2 dB steps. Effect-sizes were calculated for comparisons between the first group (centred on 5) and each of the successive groups (i.e., the mean of the first group was subtracted from the mean of each other group, and the result divided by their pooled standard deviation), for both the veridical and estimated thresholds.

To explore the effects of response consistency, we modelled "lapses" as trials where the model participant responded correctly with a probability of 0.5 (i.e., guessed) irrespective of the stimulus level (*Wichmann & Hill, 2001a*; *Wichmann & Hill, 2001b*). For the initial simulations of the effects of lapse rate on measured threshold, the model subject had a veridical threshold of $t = 10$ and slope $\beta = 1$. Thresholds were estimated with a 2-down, 1-up staircase with 1 dB steps. Lapse rate was set at 0%, 5%, or 10%. The simulations exploring effects of lapse rate on group comparisons used the same set of starting distributions of model participants as used in the group analysis described above. Lapse rates were 0%, 5% and 10% and thresholds were estimated with a 2-down, 1-up procedure using 1 dB steps. Effect-sizes for group comparisons were computed in the same way as described above.

## RESULTS

### How accurately do staircases estimate threshold in individual participants?

The first simulations explored the estimation error surrounding thresholds estimated from the staircase with the model subject having a known and fixed threshold of 10 and a known and fixed $\beta$ of 1.

Figures 2A–2C used a 2-down, 1-up staircase which theoretically converges at 70.7% correct (*Levitt, 1971*), expected to be at the threshold of 10 for this model subject. The three histograms of Fig. 2A show the effects of stopping after different numbers of reversals: 10 reversals, 20 reversals and 100 reversals. Table 2 shows the mean and standard deviation of each distribution, and indicates by how many standard deviations the veridical threshold falls below the estimated mean threshold (i.e., as a $z$-score relative to the distribution of threshold estimates).

The shape and central tendency of the distributions of estimated thresholds change as a function of the number of reversals: procedures with fewer reversals produce much broader, more kurtotic distributions. The central tendency with fewer than 100 reversals lies above the true threshold, even when estimates were based on 20 reversals. With 10 reversals the mean threshold estimate is more than 50% above the true threshold. The mean approaches the true threshold with 100 reversals, and although the distribution narrows with more reversals as the central-limit theorem would predict, the two-standard deviation range even with 100 reversals remains at ±20% of the true threshold. The fact that reversal count influences the extent to which threshold is over-estimated implies that, when comparing data between subjects or across tasks, it is very important to use the same number of reversals in each measurement.

Figure 2B and Table 2 show the results when the slope parameter, $\beta$, of the underlying psychometric function was varied. The histograms are for the same procedure as in
**Table 2  Mean and standard deviation threshold estimates. shows mean and standard deviation threshold estimates for the 2-down, 1-up adaptive procedure, under the conditions illustrated in Figs. 2A–2D and 4A.** Also shown is the $z$-score of the veridical threshold (always 10) in relation to the distribution of simulated threshold estimates. More negative $z$-scores indicate greater over-estimation of thresholds. In Fig. 2A, reversal count is manipulated for a model participant with a slope of 1, staircase step-size of 1 dB and a 2-down, 1-up adjustment rule. In Fig. 2B, the simulations are for 20 reversals with slope manipulated. Figure 2C is as for Fig. 2A except that the step-size was 2 dB. Figure 2D is as Fig. 2A except that the adjustment rule is 3-down, 1-up. Figure 4A shows data for 20 reversals as in Fig. 2A, except that lapse rate is manipulated. The asterisk indicates datasets which are identical across plots. Please refer to the figures and text for more information.

| Figure | Condition | Mean | Standard deviation | z score of veridical threshold |
|---|---|---|---|---|
| Figure 2A | 10 reversals | 16.83 | 4.55 | −1.50 |
| | 20 reversals* | 14.10 | 3.57 | −1.15 |
| | 100 reversals | 10.58 | 1.78 | −0.32 |
| Figure 2B | Slope = 0.5 | 19.29 | 6.48 | −1.43 |
| | Slope = 1.0* | 14.10 | 3.57 | −1.15 |
| | Slope = 3.0 | 10.20 | 1.25 | −0.16 |
| Figure 2C | 10 reversals | 14.17 | 5.27 | −0.79 |
| | 20 reversals | 11.97 | 3.78 | −0.52 |
| | 100 reversals | 9.81 | 1.75 | 0.11 |
| Figure 2D | 10 reversals | 11.47 | 2.66 | −0.55 |
| | 20 reversals | 10.32 | 1.90 | −0.17 |
| | 100 reversals | 9.34 | 0.89 | 0.75 |
| Figure 4A | Lapse rate = 0%* | 14.10 | 3.57 | −1.15 |
| | Lapse rate = 5% | 15.27 | 3.93 | −1.34 |
| | Lapse rate = 10% | 16.50 | 4.23 | −1.54 |

Fig. 2A with 20 reversals, and a threshold of 10. The slope was shallow ($\beta = 0.5$), $\beta = 1.0$ (as in Fig. 2A), or steep ($\beta = 3.0$). The histograms show that the procedure's tendency to overestimate threshold, and the variability of the estimates, are both greatest with shallower slopes. Therefore, knowing the slope of the underlying psychometric function would be helpful when choosing an adaptive procedure but the slope is almost never known in investigations of paediatric and/or clinical populations. A complicating factor is that in children, slope may change with age (e.g., *Buss, Hall & Grose, 2009*) and indeed potentially across different patient groups.

Step-size can also influence how quickly and well an adaptive procedure converges. In Fig. 2C, the step-size is increased from 1 dB to 2 dB for the same adaptive procedure and underlying veridical psychometric function as in Fig. 2A. The three panels again show histograms for different numbers of reversals: 10, 20 and 100. The mean threshold estimate is closer to the real threshold with 2 dB than with 1 dB steps in all three histograms, but the variance of the distribution increases slightly with increased step size (see also Table 2 for details). Although the step size can be chosen by the experimenter, its effect on threshold estimates depends on the (unknown) slope of the psychometric function underlying the task (*Levitt, 1971*). The implications of increased variance are discussed below in relation to Fig. 3.

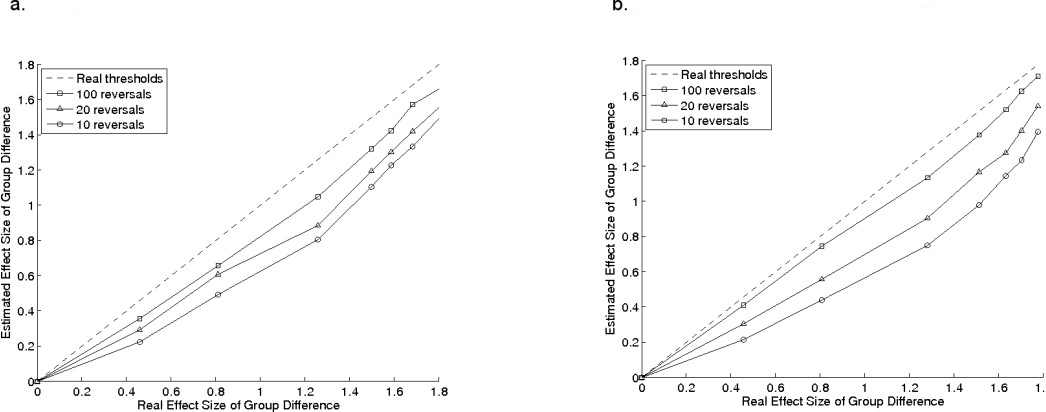

**Figure 3  Group comparisons.** Effect-sizes for group comparisons for estimated thresholds in a group of model observers, plotted as a function of the effect size for the same comparisons using their real thresholds, for a 2-down, 1-up procedure 3A and a 3-down, 1-up procedure 3B. Error bars show standard deviation.

Procedures with different adjustment rules converge at different points on the psychometric function. Figure 2D shows histograms for a 3-down, 1-up procedure which converges at 79.4% correct (*Levitt, 1971*). The model subject's veridical threshold (at 79.4%) was 10, the step size was 1 dB, and $\beta$ again was 1. The histograms of estimated threshold obtained from this simulation are slightly narrower than with the 2-down, 1-up procedure, and the central tendency of the histograms approaches the true threshold with as few as 20 reversals (see also Table 2). This improvement comes at the cost of significantly increased numbers of trials: the 2-down, 1-up staircase completed 20 reversals in an average of 67 trials ($\pm$1 standard-deviation of 8 trials) whereas the 3-down, 1-up staircase required an average of 146 ($\pm$11) trials, because it requires a longer sequence of correct responses for each downward step.

The data in Figs. 2A–2D are for a single model subject with a fixed threshold, but it is important to know if the effects shown are predictable across different thresholds. Figure 2E addresses this question using the same 2-down, 1-up procedure as in Fig. 2A, but with thresholds ranging from 1 to 20. One-thousand threshold estimates were made for each underlying (true) threshold and the mean is plotted as a function of underlying threshold. The lines are for stopping at 10 (circles), 20 (triangles) or 100 (squares) reversals. Error bars indicate $\pm$ one standard deviation and the dashed line lies on the locus of veridical estimation. The over-estimation of threshold with this procedure increases with the true threshold, and the over-estimation is greatest (and with the largest standard-deviation) for the procedure with fewest reversals. Thus comparisons of groups with different thresholds within a group will be complicated by threshold-dependent over-estimation which will increase the probability of Type 1 error. The lines in Fig. 2E become parallel on semi-log axes and, along with the bias seen in Fig. 2A, the simulations suggest that datasets obtained with adaptive procedures using logarithmic step-sizes may frequently be logarithmically skewed, thus requiring log-transformation prior to analysis.
## Effects of adaptive procedure parameters on group comparisons

For group comparisons, we created groups of 1,000 model subjects with thresholds drawn from a known normal distribution and each having underlying psychometric functions with $\beta = 1$. This approximates the scenario with samples drawn from a large inhomogeneous population, but with many more subjects than is typical.

We investigated the extent to which the number of reversals influenced the likelihood of obtaining statistically significant between-group differences. This analysis was designed to simulate a hypothetical situation where two groups of participants may differ in their average sensitivity to a stimulus. Table 3 shows the means and standard deviations of the starting distributions of veridical thresholds. The dashed line in Figs. 3A and 3B show the pairwise effect-sizes for comparisons of veridical thresholds, and the remaining lines show effect-sizes for the comparisons obtained from adaptive procedures with 10 (circles), 20 (triangles), and 100 (squares) reversals, with a 2-down, 1-up procedure with a 1 dB step size (Fig. 3A), and the 3-down, 1-up procedure with 2 dB steps (Fig. 3B). In both cases, the effect-size of the comparison for estimated thresholds is smaller than it would be for the real thresholds, and is smallest when fewest reversals are used. This has implications for researchers comparing groups of children; the smaller the effect-size, the less the likelihood of detecting a real difference between the groups with standard statistical tests. It follows that if fewer reversals are used, larger groups of participants are needed to detect group differences. Table 3 shows the means and standard deviations for the estimated thresholds and also the number of participants that would be required to find a statistically significant difference between the first group and each of the successive groups in a 2-sample $t$-test (see legend for details). Even with 100 reversals, nearly twice as many participants are needed to detect a difference between the first and second groups as for the veridical thresholds. For 10 reversals, four times as many are required.

## Effects of response consistency on individual thresholds and group comparisons

Our simulations so far have assumed that subjects perform consistently; i.e., the probability of making a correct response is determined entirely by their underlying psychometric function. However such consistency is unlikely for real participants—even if they are highly trained and highly motivated, so the following simulations explore the effects of differing lapse rates.

Figure 4A shows histograms of data from 2-up 1-down staircases for three different lapse rates, with threshold estimation based on 20 reversals for an underlying psychometric function with a threshold of 10 and $\beta = 1$. The data for the lapse rate of 0 is the same as in the 20 reversals section of Fig. 2A and shows results when there are no lapses. Data from *Hulslander et al. (2004)*, from children with dyslexia, suggest a catch trial failure rate of 5–10%. As the lapse rate increases from 5% to 10% the central tendency of the histogram shifts farther from the true threshold, but the relative spread of the distribution remains roughly constant at 1.8 times the mean. See also Table 2; note that a 10% lapse rate is only half that found on average in children by *Talcott et al. (2002)*. Because the standard deviation of estimated thresholds with changing lapse rate is proportional to the mean

Witton et al. (2017), *PeerJ*, DOI 10.7717/peerj.3231

**Table 3 Group comparison data from Fig. 3A—statistics.** Statistics for the group comparison data in Fig. 3A. Means and standard deviations ('*s.d.*') are given for the distributions with each nominal mean value between 5 and 12 (left column), for the randomly-generated starting distributions of real thresholds, and for the estimated thresholds from 2-down, 1-up (1 dB) staircases with 10, 20, and 100 reversals. Also shown for each set of distributions are the required numbers of cases ('*req. n*') for a statistically significant group difference when compared with the first distribution (centred on 5), based on a two-sample $t$-test with alpha level of 0.05 and 80% power.

| Nominal mean value | Starting distributions | | | Staircase: 10 reversals | | | Staircase: 20 reversals | | | Staircase: 100 reversals | | |
|---|---|---|---|---|---|---|---|---|---|---|---|---|
| | *mean* | *s.d.* | *req. n* | *mean* | *s.d.* | *req. n* | *mean* | *s.d.* | *req. n* | *mean* | *s.d.* | *req. n* |
| 5 | 4.97 | 0.99 | . | 8.51 | 2.86 | . | 7.02 | 2.23 | . | 5.28 | 1.38 | . |
| 5.5 | 5.48 | 1.12 | 38 | 9.18 | 3.10 | 160 | 7.73 | 2.57 | 94 | 5.81 | 1.55 | 64 |
| 6 | 6.00 | 1.24 | 14 | 10.16 | 3.60 | 35 | 8.63 | 2.78 | 24 | 6.34 | 1.66 | 21 |
| 7 | 6.94 | 1.46 | 8 | 11.64 | 4.15 | 15 | 9.87 | 3.41 | 13 | 7.38 | 1.98 | 10 |
| 8 | 8.01 | 1.56 | 6 | 13.58 | 4.60 | 9 | 11.37 | 3.48 | 8 | 8.49 | 2.18 | 7 |
| 9 | 8.99 | 1.91 | 6 | 14.95 | 5.12 | 8 | 12.75 | 4.16 | 7 | 9.52 | 2.61 | 7 |
| 10 | 10.02 | 1.98 | 5 | 16.58 | 5.70 | 7 | 14.28 | 4.58 | 7 | 10.67 | 2.67 | 6 |
| 12 | 11.92 | 2.25 | 5 | 20.41 | 6.92 | 6 | 17.17 | 5.34 | 6 | 12.64 | 3.20 | 6 |
a.

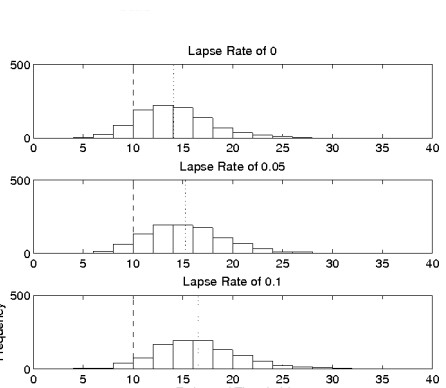

b.

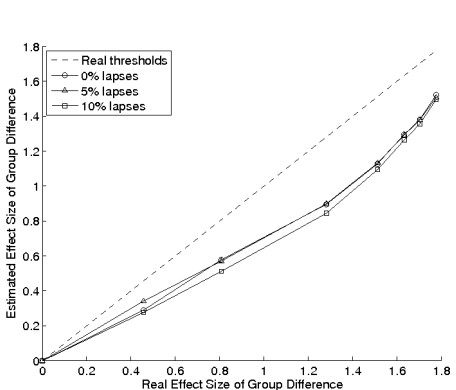

c.

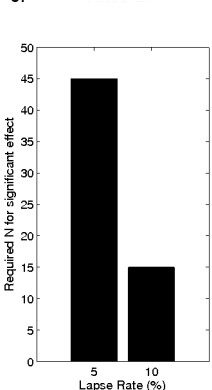

**Figure 4** **The effects of lapse rate on estimated threshold.** (A) shows histograms of estimated thresholds, taken from 20 reversals, for a single model observer with a real threshold of 10 ($\beta = 1$), with different lapse rates. The data for the lapse rate of 0% in (A) are the same data as in the 20 reversals section of Fig. 2A. (B) shows the effect of lapse rate on mean estimated threshold across the same groups of model observers as in the reversal-count analysis from Fig. 3. (C) illustrates the group-sizes that would generate an *artificial* group difference for groups with lapse rates of 5% or 10%, even when veridical thresholds in both groups were identical, using the data in (A).

estimate, the effect-size of between-group comparison does not depend on lapse rate. This is shown in Fig. 4B, which uses the same sample-distributions as in Fig. 3. The effect-sizes of the group-comparisons derived from estimated thresholds are lower than those obtained for the real thresholds, but the magnitude of this reduction does not depend systematically on lapse rates. Importantly, this result relies on lapse rates being the same in all groups. Significant problems in the form of increased risk of Type 1 error rate will emerge if lapse rates differ between groups, as may be the case when comparing normal and clinical groups of children (see *Hulslander et al., 2004*, for example data). In other words, given two observers with equal veridical thresholds but different lapse rates, the estimated threshold for the observer with the higher lapse rate will be drawn from a probability distribution with a higher mean value. Thus groups of observers with higher lapse rates will exhibit higher thresholds than a group with lower lapse rates even if their veridical thresholds are similar,

an effect which artificially increases the effect-size for the between-group differences. To illustrate this problem, we used the same approach as in Table 3 to compute the number of participants needed for a significant group difference in a $t$-test (with an alpha of 0.5 and 80% power), using the data in Fig. 4A, where all groups which have the *same veridical threshold* of 10 but different lapse rates. Compared to the group making 0% lapses, the group making 5% lapses would show an artificial, statistically significant, group difference if they contained 45 individuals (Fig. 4C). A significant (and false) group difference would emerge with only 15 individuals if the second group were making lapses on 10% of trials (2-sample $t$-test, 80% power, $p < 0.05$). The implications of this are clear for researchers comparing groups which may differ in lapse rate.

## DISCUSSION

Adaptive psychophysical procedures were designed for use in trained observers where the psychophysical properties of a task are well-defined, but they are also widely used to measure sensory thresholds in studies of untrained adults, children, and clinical populations. Measurements are often based on relatively small amounts of data in order to minimise number of trials, and hence reduce the risk of poor motivation or unreliable performance. Consistent responding is a particular challenge when working with children because of developmental factors. For example, compared to adults, children often have difficulties maintaining attention during even the shortened series of trials required an adaptive procedure; and those with limited attentional control, such as children with attention-deficit disorder (ADD) may be even more likely to lose vigilance. The simulations presented here illustrate the problems of reducing the number of trials, and hence increasing the effects of inconsistent responding on thresholds estimated from adaptive procedures. They draw on one commonly-used staircase method to illustrate the problems that can arise when measuring thresholds, and comparing across groups of individuals. The results show that adaptive procedures can over-estimate thresholds, and that this tendency is greater when fewer reversals are used. This introduces experimental error into threshold estimation, making it harder to detect group differences, and hence increasing in the likelihood of Type 2 errors, in failing to reject the null hypothesis.

The results also showed increased bias towards over-estimation of higher thresholds, i.e., the higher the veridical threshold, the greater the bias in its estimation by the adaptive procedure. This asymmetric bias increases the possibility that data-sets arising from multiple adaptive procedure measurements will not be normally distributed, although this trend may not be detected with small numbers of participants. Finally, observers' lapse rates also influence measured thresholds by shifting the estimated thresholds further from the true threshold as the probability of lapses increases. Differences in lapse rates between groups significantly influence the effect-size of a group comparison. This could lead to apparent group differences when there are no differences in underlying thresholds (i.e., Type 1 errors).
## How many reversals?

In this study, the measurement error associated with a psychophysical threshold (i.e., the standard deviation of threshold estimates in our simulations) depends strongly on the number of reversals (Figs. 2 and 3). Specifically, the variability of threshold estimates is larger when fewer reversals were used. When working with children or untrained participants, researchers can typically only draw one (or maybe a handful) of estimates from this probability distribution for each individual, making it difficult to know whether the measured value is from a point close to the mean or in one of the tails. So, with small numbers of reversals, care must be taken when comparing individual thresholds. The other important implication of using fewer reversals is that group comparisons have reduced statistical power (Fig. 3) for any given group-size.

Unfortunately, it is impossible to recommend an ideal number of reversals to achieve an acceptable level of accuracy at the individual level, or adequate statistical power for group statistics. This is because the distributions of thresholds are a product of interactions between the (often unknown) slope of the psychometric function underlying any given task, and the adjustment rule and step-size of the adaptive procedure. Researchers might consider using information from the literature, or, better, from detailed pilot measurements of full psychometric functions in a small sample of their own subjects, to determine which adaptive procedure might be most efficient for a given task. Ultimately, maximising the number of reversals as far as possible is key to obtaining more accurate estimates, and the use of a procedure which converges at a higher point on the psychometric function, such as the 3-down, 1-up procedure, is also likely to be helpful. For example, *Buss et al. (2001)* explored the accuracy of adaptive procedures using a 3-down, 1-up staircase with 2 dB steps in normal 6–11 year old children and obtained auditory detection thresholds that they accepted as stable based on a relatively small number of reversals. The challenge for researchers is the risk that running a longer adaptive procedure (such as the 3-down, 1-up procedure—which required more than double the number of trials in our simulations) could result in a higher lapse rate, which brings the additional problems discussed in detail below. Finally, we note the critical importance of using the same number of reversals for the measurement with each participant in a study. This is because the extent to which threshold is typically over-estimated depends on the number of reversals—thresholds for different reversal counts are therefore not comparable.

## Lapses and how to handle them

The most significant problem associated with lapses on a psychophysical task is that they are impossible to measure—in practice, incorrect responses that result because the participant was not attending to the stimulus are not possible to detect from the data alone. Nevertheless, the psychometric function might hold some information about the lapse rate: always assuming that lapse rate is approximately independent of stimulus level, its upper asymptote will be reduced from 100% correct by half the lapse rate. For example, at a lapse rate of 5%, the psychometric function will asymptote at 97.5%. *Wichmann & Hill (2001a)* included lapse rate as a free parameter in their fitting procedure for psychometric functions (though not as a parameter of the function itself), to preclude estimates of

threshold and slope from being severely affected by trained observers failing to reach 100% correct responses. Thus asymptotic performance can be used to estimate the lapse rate to obtain better estimates of the true thresholds. Adaptive procedures, however, do not typically contain information about the upper asymptote of the psychometric function, and while lapse rate and slopes can be estimated from certain adaptive procedures, they interact (*Wichmann & Hill, 2001a*; *Wichmann & Hill, 2001b*).

An alternative strategy for estimating lapse rate is to use "catch trials"; a fixed proportion of trials, not contributing to stimulus level adjustments, but where the stimulus level is set at a value sufficiently high to lie on the upper asymptote of the underlying psychometric function. Assuming that lapses are independent of the stimulus level, the performance on these catch trials provides some estimate of the lapse rate. Catch trial performance has been used successfully as a covariate in multivariate studies of reading disorder and auditory processing (for e.g., *Talcott et al., 2002*; *Hulslander et al., 2004*).

There are two potential problems with catch trials. First, when occurring unexpectedly in a sequence of near-threshold trials, they may appear unusual, attract the attention of the subject, elicit a different response for that trial, and not really reflect true lapse rate. Second, the interpretation of catch trials depends on the assumption that lapses are independent both of stimulus level and position in the measurement run. *Leek, Hanna & Marshall (1991)* successfully found a way to estimate lapse rates, without the assumption that they have constant probability, based on pre-computed confidence intervals for a pair of simultaneously-operating staircases.

Another possibility, which has been used in the literature, is to run two threshold measurements and check for consistency between their results using correlational methods. The potential problem with this approach is that one longer staircase is generally better than the average of two shorter ones. Although the total number of reversals may be the same, the bias (and hence risk of Type 1 error) and the measurement error (associated with risk of Type 2 error) are both lower when the longer staircase is used. Running another simulation of the subject from Fig. 4A, a single adaptive procedure with more reversals yields a lower threshold than an average of two shorter ones, even when lapses were being made. The average of two simulated procedures with 10 reversals each was 17.1, 18.1, and 19.2 for lapse rates of 0, 5% and 10%, respectively; whereas the procedures with 20 reversals yielded mean thresholds of 14.3, 15.4, and 16.3. The bivariate correlations between individuals' thresholds from consecutive runs for groups of observers also fail to yield sufficient information about lapse rate. For example, in a distribution of 1,000 simulated observers with a mean threshold of 10 and standard deviation of 2.5, correlations between pairs of thresholds obtained with 100 reversals each are relatively stable, at 0.67, 0.7, and 0.65 for our 3 lapse rate conditions. This stability in correlations across differing lapse rates happens because lapses alter the mean of the probability distribution of thresholds, but not its relative standard deviation.

Checking for consistency of reversal points within a staircase run is another intuitive potential approach to identifying data with lapses. However in the same simulation for 20 reversals, the standard deviation of reversal points was 5.2, 5.4 and 5.4, respectively, providing no information about the presence of the lapses. This probably happens because

the range of reversal points is not extended by these lapses but is simply shifted (0% lapses, mean range 9.9–27.2; 5% lapses, 9.9–28.4; 10% lapses, 10.6–29.4). It is worth noting that the lapse rates tested here are purposefully conservative and probably don't represent the poorest performance that is observed in some studies with children (see, for example, the plots in *McArthur & Hogben, 2012*). If a participant lapses consistently over a long period during a run of trials, for example, then the effects of this may be visible in the measures tested above. However, these measures clearly do not identify participants who lapse randomly at low rates, despite the impact that these lapses have on the measured threshold estimate.

The problem of lapses in psychophysical data is therefore difficult to solve in a satisfying way. An alternative approach to measuring sensory sensitivity, which bypasses the need for obtaining behavioural response from participants, is to use neurophysiological measures. Mismatched negativity (MMN) is an evoked response elicited by a change in a stimulus parameter embedded in a sequence, and which has been used to index sensory sensitivity in a range of developmental settings (*Näätänen et al., 2007*). The MMN response is modifiable by contributions from sources in the frontal lobes, and is sensitive to the cognitive symptoms of disorders such as schizophrenia, so although considered pre-attentive in origin it is not entirely free of cognitive influence. *Bishop (2007)* has provided a critical review of the use of this method in research of developmental disorders. It is also possible to construct "cortical psychometric functions" from auditory evoked responses measured with neurophysiological data, a method which shows promise for bias-free estimates of threshold (*Witton et al., 2012*). Yet there are challenges associated with using neuroimaging techniques with children (*Witton, Furlong & Seri, 2013*) and for the majority of studies, psychophysics will remain the method of choice. Developing strategies to reduce the likelihood of lapses during adaptive procedures, especially through improving task engagement by children (e.g., *Abramov et al., 1984*), is therefore critical—as is the use of statistical methods which are sensitive to the limitations of these procedures.

Future behavioural studies taking an individual-differences approach (e.g., *Talcott, Witton & Stein, 2013*) can potentially help improve our understanding of the link between cognitive factors such as attention and memory, and psychophysical performance, especially if these studies make detailed estimates of psychometric functions and lapse rates. Convergent measures, especially physiological measures such as eye-movement recordings, which can monitor a child's physical engagement with a visual stimulus, would also improve the extent to which researchers can determine the validity of individual trials. Finally, the application of neuroimaging techniques, especially those with high temporal resolution (i.e., MEG and EEG) could provide useful evidence to help unpick the cognitive processes that underpin variable task performance.

## CONCLUSIONS

Overall, the findings from the simulations presented here suggest that the accuracy and efficiency of studies using adaptive procedures in untrained and especially paediatric populations are best maximised by very careful choice of adaptive procedure, taking into

account the psychophysical properties of the task and stimulus. This should be followed by careful statistical analysis, especially when comparing groups. Investing in innovations able to improve quality time-on-task, particularly for children, in relevant studies will greatly improve data quality, if trial-numbers can be increased. Attempting to index individuals' lapse rates, and incorporating this information into statistical analyses, would also enable researchers to account for the impact of such differences on experimental findings.

## ACKNOWLEDGEMENTS

The authors would like to thank Professor Genevieve McArthur, Dr. Nicholas Badcock, and an anonymous reviewer for their insightful comments on an earlier version of this paper.

### Funding
The authors received no funding for this work.

### Competing Interests
The authors declare there are no competing interests.

### Author Contributions
- Caroline Witton conceived and designed the experiments, performed the experiments, analyzed the data, contributed reagents/materials/analysis tools, wrote the paper, prepared figures and/or tables, reviewed drafts of the paper.
- Joel B. Talcott conceived and designed the experiments, wrote the paper, reviewed drafts of the paper.
- G. Bruce Henning conceived and designed the experiments, analyzed the data, contributed reagents/materials/analysis tools, wrote the paper, reviewed drafts of the paper.

### Data Availability
  The code has been supplied as a Supplementary File.

### Supplemental Information
Supplemental information for this article can be found online at http://dx.doi.org/10.7717/peerj.3231#supplemental-information.

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
