# Peer review of "Psychophysical measurements in children: challenges, pitfalls, and considerations"

_PeerJ, doi:10.7717/peerj.3231_

## Round 0.1 · original submission · Minor Revisions

Dear Caroline,

Thank you for sending your manuscript to PeerJ for expert review. Although "importance" and "novelty" are not criteria for publication in PeerJ, I agree with two expert reviewers that the topic you address in your manuscript is indeed important. I also agree that the methods that you used to address this issue were appropriate in general, and that the reporting of your study was clear overall.

That said, as is always the case, there are a number of places in the manuscript where clarity could be improved. In addition, one reviewer requests evidence for statistical reliability, which I agree would further strengthen the outcomes of the study. Thus, as academic editor, I recommend that you address each of the comments made by the reviewers in order to strengthen the manuscript - ideally in the manuscript or, if not possible, in a response to the reviewers. They have provided some very clear suggestions about how you might address their concerns and hence I predict a revision will not prove too arduous.

I look forward to seeing a revised version of your interesting manuscript in due course. I hope you have a very happy new year.

Best wishes,

Genevieve

Reviewer 1 ·

Basic reporting

The report is well written with clear and professional English language used throughout. The introduction introduces lots of key terminology which might be better presented in a table format to enable the less knowledgeable reader to refer to more easily.

The authors describe adaptations to the typical psychophysical methods which have been used in the adult literature to make tasks suitable for use with children – it would be useful if the authors cited some examples of studies which have used tasks adapted in this way.

The figures are relevant and clear however the interpretation of figures 2a-2e & 4a would be aided if notes regarding the values used for key factors (e.g. value of slope, threshold, step size, n. reversals etc) in the simulations were included in figure tiles/notes. This would enable the reader to make comparisons between figures without having to refer to the main text to find this information. The authors could then just focus on describing and interpreting the key findings in the main text.

It might be useful to include some illustrations of a psychometric function and/or adaptive procedure obtained with a child to illustrate how performance can differ e.g. variability in responding, shallow slopes, higher thresholds etc.

Minor points:
Line 171 check heading font (here and throughout paper)
Iine 191 use of comma and semi-colon ,;
Line 270 refers to footnote 1. Should there be an additional footnote in the paper?
Line 412 delete line

Experimental design

The method section seems quite sparse and a little difficult to follow. Can the authors provide details of how the number of reversals, the adjustment rule and response consistency were manipulated? i.e. the different levels used. It would be helpful to perhaps include details of the different factors (e.g. value of slope, threshold, step size, reversals, adjustment rule, n. trials etc) you consider in the simulations here and how you plan to test factors (and what these factors are) that might affect group comparisons. If some of this information was moved to the method the results would be less dense and the authors could focus more on describing key findings.

Validity of the findings

The authors summarise the results well in the discussion and highlight key things to consider when using psychophysical tasks with children and suggest alternative options too. The best way forward regarding using these types of tasks with children and clinical groups is still somewhat illusive and I think there is room here for the authors to suggest further research that needs to be done.

Could the authors comment on the utility of reporting measures of threshold stability e.g. standard deviation across final reversals (McAuthur and Hogben, 2012) alongside simple threshold estimates as an estimate of reliability. These measures maybe particularly useful when interpreting results in developmental samples.

The current simulations have been conducted based on a 2AFC task however an alternative task which is often used with children is an AXB ‘odd-one-out’ paradigm which reduces the need for children to understand concepts like ‘higher’, ‘lower’, ‘longer’, ‘shorter’ etc. Can the authors comment on whether their results speak to these types of tasks? Or whether task type is likely to influence lapse rate?

Additional comments

The study aims to address an important and often overlooked issue; namely the validity of using psychophysical measurements in children. To test causal theories of developmental disorders (e.g. Dyslexia, language disorder and ADHD) which implicate lower-level sensory deficits psychophysical methods are often used. Results in the literature are often inconsistent and across studies different methods have been used to adapt these types of tasks for use with children. A common practice is to shorten tasks by using adaptive procedures with a reduced number of reversals however children’s responses are still often inconsistent. Testing the impact of these factors using simulations is an important step towards understanding how task adaptations and inconsistent responding impacts on threshold estimates and whether the resulting estimates are reliable and valid. This study provides an important addition to the field.

·

Basic reporting

The work is clearly written and well presented. Consideration of the following thoughts/comments may increase the transparency of the work.
1. Page 8 line 74/75: “Each of these factors has a normal developmental trajectory, but that trajectory can be significantly disrupted by the presence of a developmental disorder.” > Useful to include an example reference for this
Regarding Figure 1: “In Figure 1b, the procedure terminates after 20 reversals, two of which are indicated by circles.”
2. Quite possible that all of the reversals aren’t visible but I only counted 14. Would be good to make this very clear for the naïve reader – possibly highlight each reversal with a filled dot
3. Also unclear why those 2 reversal points are highlighted – consider highlighting all and/or making the final two stand out rather than somewhere in the middle (unless there’s a rationale for these 2 points that I have missed)

4. Figure 2e > Consider horizontally offsetting each reversal condition for clarity of error bar overlap – quite hard to discern with the horizontal alignment
5. On Page 14, line 212: “…tracking rule of the staircase…” > Please considering providing an example of the tracking rule at this point for clarity
6. Regarding Figures 3 a and b: Consider setting the x and y axes to be constant between panels a and b
7. On Page 21 lines: 372-378 – lapse rate comparisons: Please consider including a figure or table to clarify this comparison

Experimental design

The work is well motivated and very useful for people employing psychophysical procedures with children and special populations. The design is sound and well conducted. Relevant MATLAB scripts are provided to recreate the simulated data and figures. I’ve tested the Figures.m function and it produces the figures and therefore the data very nicely.

Validity of the findings

Regarding Page 16, line 245: “The mean approaches the true threshold with 100 reversals…”
8. Although there’s visually no question about these results, it seems useful to report the means and effect sizes to give readers the magnitude of the error we’re looking at. Suggest doing this for all of the comparisons.
9. Consider including one-sample t-tests against the threshold (i.e., 10)
10. I note that the means and SD appear on page 17 in comparison to the step size analysis – consider placing these within a table and referring to for the 2a analysis.
11. On a related note, following the means on page 17, there’s an indication of a footnote (super text 1) but I can’t find the footnote. Please check.

Additional comments

Overview:
This paper presents a series of simulations testing the influence of a number of parameters on threshold estimates derived from adaptive psychophysical procedures. The simulations provide a tidy introduction to the threshold estimates due to variations in the number of reversals, slope of the psychometric curve, step size, and decision rule, as well as variation due to threshold value, group differences, and lapses in attention.

Dear Caroline and colleagues,

This is a well written piece of work and provide a useful and practical guide to working with adaptive psychophysical procedures with children and special populations. I have one main suggestion about reporting the statistical reliability of the results and a few specific comments that may increase the transparency of some of explanations and figures.

Best wishes,
Nic Badcock

---

## Round 0.2 · Minor Revisions

Dear Caroline (BTW, this is the real me here, not the automated me),

Thanks for your revised version of your manuscript. I have looked at your responses to the reviewers' suggestions, and your changes to the manuscript, and I believe that you have satisfied their concerns or suggestions, so I will not be sending it back to them for re-review.

Since we now have a solid foundation for the study in place, I re-read the manuscript with a strong focus on readability. As is, the manuscript would be fine for people who are very familiar with using psychophysics with children (as the reviewers, and I, are). However, we would really like to make your paper as easy to understand as possible for people who are planning to do this for the first time.

So, I have been through the manuscript with a fine-tooth comb. I have attached a pdf copy of my suggested changes, which relate mostly to wording, and also some shifting of background information from Results (which seems a bit odd) into the Methods section, and formatting. I don't think I can send you a word version of this file through the PeerJ system (ie so you can just accept or reject these suggested changes). If you would like a copy of this file, just shoot me an email.

Once these final suggestions/issues have been addressed, I will be able to accept the manuscript, and then you and copy editors can do their double checks for final wording/formatting issues during the proofing process.

Best wishes,

Genevieve

---

## Round 0.3 · accepted · Accept

Dear Caroline,

It's the real me here. Good work on all the revisions. I am going to accept the manuscript so that we can keep moving forward towards publication. However, when you get the proofs, can you keep an eye out for the following things to make sure they are correct:

1. In one of the two abstracts supplied, there is still the use of Type-1 and Type-2.
2. In Abstract, change "group-differences" to "group differences"
3. On line 68, ensure there is no gap in "7- 12 years"
4. One lines 75-76, ensure references are in alphabetical order
5. Use "2AFC" rather than "2-AFC" throughout manuscript
6. On line 199, change double negative (not unknown) to "possible"
7. In a couple of places, there are entire sentences in brackets (lines 239 and 281 and 376). Either integrate into previous sentence (in brackets) or transform into proper sentences.
8.Line 235 - change ";" to ","
9. Line 239 - merge info in two separate brackets into one using ";"
10. Line 295 - fix ".."
11. Line 375, add "," after "dyslexia"
12. Line 429 - "depend" should be "depends"?
13. Change "catch-trial" to "catch trial" throughout
14. Ditto for "lapse rate".
15. Line 510 - change "for example" to ",for example,"
16. Insert ,"" after "recordings"
17. Line 540 - put "i.e., MEG and EEG" in brackets
18. First sentence of Conclusions is very long. Break up into two sentences.

Best wishes,

Genevieve